

# Patterned progression of gut microbiota associated with necrotizing enterocolitis and late onset sepsis in preterm infants: a prospective study in a Chinese neonatal intensive care unit

Jiayi Liu[1], Yuqing Li[1], Yi Feng[1], Liya Pan[1], Zhoulonglong Xie[2], Zhilong Yan[2], Li Zhang[3], Mingkun Li[3,4], Jianhua Zhao[5], Jianhua Sun[6] and Li Hong[1]

[1] Department of Clinical Nutrition, Shanghai Children's Medical Center, School of Medicine Shanghai Jiao Tong University, Shanghai, China
[2] Department of Surgery, Shanghai Children's Medical Center, School of Medicine Shanghai Jiao Tong University, Shanghai, China
[3] Key Laboratory of Genomic and Precision Medicine, Beijing Institute of Genomics, Chinese Academy of Sciences, Beijing, China
[4] Center for Excellence in Animal Evolution and Genetics, Chinese Academy of Sciences, Kunming, China
[5] Shanghai Majorbio Bio-pharm Technology Co., Ltd, Shanghai, China
[6] Department of Neonatology, Shanghai Children's Medical Center, School of Medicine Shanghai Jiao Tong University, Shanghai, China

Corresponding authors
Jianhua Sun,
sunjianhua@scmc.com.cn
Li Hong, hongli@scmc.com.cn

## ABSTRACT

Necrotizing enterocolitis (NEC) and late-onset sepsis (LOS) are two common premature birth complications with high morbidity and mortality. Recent studies in Europe and America have linked gut microbiota dysbiosis to their etiology. However, similar studies in Asian populations remain scant. In this pilot study, we profiled gut microbiota of 24 Chinese preterm infants from birth till death or discharge from NICU. Four of them developed NEC and three developed LOS. Unexpectedly, we detected highly-diversified microbiota with similar compositions in all patients shortly after birth. However, as patients aged, the microbial diversities in case groups differed significantly from that of the control group. These differences emerged after the third day of life and persisted throughout the course of both NEC and LOS. Using a Zero-Inflated Beta Regression Model with Random Effects (ZIBR), we detected higher *Bacillus* ($p = 0.032$) and *Solibacillus* ($p = 0.047$) before the onset of NEC and LOS. During NEC progression, *Enterococcus*, Streptococcus and *Peptoclostridium* were the dominant genera while during LOS progression; *Klebsiella* was the only dominant genus that was also detected by the diagnostic hemoculture. These results warrant further studies to identify causative microbial patterns and underlying mechanisms.

## INTRODUCTION

The gut microbiota is a crucial contributor to human health. Imbalance of the microbial community, termed dysbiosis, is associated with various diseases, such as obesity and diabetes (*Bouter et al., 2017*; *Rosenbaum, Knight & Leibel, 2015*; *Winer et al., 2016*; *Cani, 2019*; *Zmora, Suez & Elinav, 2019*), immunity-related diseases (*Vogelzang et al., 2018*; *Pronovost & Hsiao, 2019*; *Vatanen et al., 2016*), neurodevelopmental disorders (*Sampson & Mazmanian, 2015*; *Pronovost & Hsiao, 2019*), cardiovascular diseases (*Tang, Kitai & Hazen, 2017*; *Jie et al., 2017*; *Jonsson & Bckhed, 2017*) and cancers (*Gagliani et al., 2014*; *Irrazábal et al., 2014*; *Sears & Garrett, 2014*).

The microbiota in newborn infants undergoes dynamic changes in composition, abundance and diversity before reaching homeostasis at around three years of age (*Yatsunenko et al., 2012*; *Bäckhed et al., 2015*; *Stewart et al., 2018*). Temporal colonization pattern of the intestinal microbiota during early stages of life may have an important contribution to the long term health of an individual. Early life microbiota disruption had been associated with the development of metabolic and immunological diseases such as Type I diabetes (*Giongo et al., 2011*; *Vatanen et al., 2018*), asthma (*Stokholm et al., 2018*) and allergies (*Madan et al., 2012a*; *Savage et al., 2018*).

In preterm infants, common medical practices including Cesarean delivery, formula feeding, sterile incubator nursing and extensive use of broad-spectrum antibiotics may limit the normal microbiota acquisition and development (*La Rosa et al., 2014*; *Shin et al., 2015*; *Deweerdt, 2018*). Resultant abnormal microbiota colonization in the gut may then contribute complications such as necrotizing enterocolitis (NEC) and late onset sepsis (LOS) (*Sharon et al., 2015*; *Cernada et al., 2016*).

Necrotizing enterocolitis is characterized by rapid ischemic necrosis of intestinal mucosa, resulting in high morbidity (2%–7%) and mortality (15%–30%) (*Neu & Walker, 2011*; *Stoll et al., 2015*). Its etiology remains largely unknown and likely to be multi-factorial. Previous studies in European and American countries have associated microbial dysbiosis to NEC onset. Reduction in microbiota diversity and unusual species colonization were observed in NEC patients (*Jacquot et al., 2011*; *Warner et al., 2016*). No causative species have been identified so far. However, an increase in Proteobacteria phyla and a decrease in Firmicutes were observed before NEC onset (*Mai et al., 2011*; *Zhou et al., 2015*). Besides, blooming of *Gammaproteobacteria* and under-representation of *Negativicutes* were associated with disease progression (*Warner et al., 2016*).

Late onset sepsis (LOS) is another common life-threatening disease for preterm infants. It is commonly defined as a systemic infection with the isolation of pathogenic bacteria from the bloodstream after 72 h of life (*Rao, Ahmed & Hagan, 2016*; *Pickering, Baker & Kimberlin, 2012*). Preterm infants have immature gastrointestinal and immune systems. Therefore, it is easier for pathogenic bacteria or bacterial toxins that can cause systemic inflammation to enter the bloodstream (*Schwiertz et al., 2003*; *Bezirtzoglou, Tsiotsias & Welling, 2011*; *Cernada et al., 2016*; *Sharon et al., 2015*; *Korpela et al., 2018*), thus making the intestine a potential source of infections and inflammation. Previous studies showed that the LOS patients' gut microbiota was less diversified, and dominated by *Staphylococci*

and *Enterobacter* but underrepresented by probiotic *Bifidobacteria* (*Madan et al., 2012b*; *Tarr & Warner, 2016*; *Stewart et al., 2017*; *Korpela et al., 2018*; *Ficara et al., 2018*).

NEC and LOS are two major causes of morbidity and mortality in preterm infants worldwide and have been exerting economic burdens on healthcare costs (*Johnson et al., 2013*; *Johnson et al., 2014*; *Mowitz, Dukhovny & Zupancic, 2018*). Although early recognition and treatment regimen has improved clinical outcomes, both diseases still account for morbidities in NICU survivors (*Hintz et al., 2005*; *Zonnenberg et al., 2019*; *Shah et al., 2015*). In China, the rate of preterm birth is as high as 7.1% (*Blencowe et al., 2012*) and continuous improvements in neonatal health care have greatly improved the survival of preterm infants. However, the risk of developing NEC and LOS increases as well. Elucidating their pathogenesis and developing preventive strategies would greatly benefit the health of preterm infants. Motivated by this, we carried out this longitudinal pilot study to profile the microbiota of Chinese preterm NEC and LOS patients, with the aim to examine if similar alterations in microbiota correlate with the onset and progression among Chinese patients. Consistent with previous studies in Western countries, we observed lower bacterial diversity among Chinese NEC and LOS patients. In contrast, we found that the Chinese patients in our cohort showed different bacterial compositions.

## METHODS

### Ethics

This study was approved by the joint committee of ethics of Shanghai Children's Medical Center, Shanghai Jiao Tong University School of Medicine (SCMCIRB-K2013022). Detailed written informed consent was obtained from parents before enrolment.

### Patients

Newly born preterm infants with a gestational age less than 33 weeks and birth weight over 950g were enrolled from Neonatal Intensive Care Unit (NICU) at Shanghai Children's Medical Center from July 2013 to December 2014. The exclusion criteria were (1) diagnosed with early-onset sepsis, (2) hepatic diseases, (3) renal impairment (Cr > 88 $\mu$M), 4) diagnosed with intestinal obstruction, (5) in foreseeable need of major cardiovascular or abdominal surgeries (except for male circumcision or PDA ligation), (6) estimated parenteral support to supply over 50% of daily caloric intake for more than four days, (7) given intravenous antibiotics administration (except prophylactic regimen of cefotaxime, piperacillin-tazobactam and/or metronidazole), (8) history of oral antibiotics administration, (9) grossly bloody stools at admission, and (10) over five days old.

NEC cases were defined as infants who met the criteria for Stage II and Stage III NEC diagnosis (*Bell et al., 1978*), including radiographic intestinal dilation, ileus, pneumatosis intestinalis, and/or absent bowel sounds with or without abdominal tenderness, and/or mild metabolic acidosis and thrombocytopenia. An LOS case was defined if an infant (1) had a positive hemoculture or other suspicious loci of infection after 72 h of life, or (2) presented with septic signs/symptoms reviewed and diagnosed independently by at least two neonatologists, and had been responding well with advanced antibiotics (e.g., Meropenem) after diagnosis. Infants with no infectious complications were regarded as controls.

## Sample collection and handling

Fecal sample collection started from neonatal meconium until death or discharge, whichever came first. Although we intended to collect fecal samples every day, due to working shifts and flexible clinical scheduling, we set seven days as the maximum interval between two collections from every infant. Every sample was collected from infants' diaper with a sterile spatula into cryogenic vials within 30 min of defecation. Then the sample was immediately placed on dry ice and stored at −80 °C within 30 min without additives. All samples were collected and stored before knowing the diagnosis of respective patients.

## DNA extraction and quality control amplification and 16s rRNA gene sequencing

Microbial genomic DNA was isolated from each fecal specimen using the E.Z.N.A. Soil DNA Kit (Omega Bio-Tek, Norcross, GA, U.S.) according to the manufacturer's protocols. The concentration and purity of the DNA were determined by NanoDrop 2000 UV-vis spectrophotometer (Thermo Scientific, Wilmington, DE, USA), and the DNA quality was checked by 1% agarose gel electrophoresis.

## Broad-range PCR and High-throughput Sequencing of 16s rRNA gene amplicons

The V3-V4 hypervariable regions of the bacterial 16S rRNA gene were amplified by PCR from each sample using bacterialarchaeal primers 338F (5′-ACTCCTACGGGAGGCAGCAG-3′) and 806R (5′-GGACTACHVGG GTWTCTAAT-3′) using a thermocycler PCR system (GeneAmp 9700, ABI, USA). The PCR reactions were as follows: 3 min of denaturation at 95 °C, 27 cycles of 30 s at 95 °C, 30 s annealing at 55 °C and 45 s elongation at 72 °C, and a final extension at 72 °C for 10 min. The PCR reactions were performed in triplicate, with each 20 μL mixture containing four μL 5X FastPfu Buffer, two μL 2.5 mM dNTPs, 0.8 μL of each primer (five μM), 0.4 μL FastPfu Polymerase (TransGen Biotech, Beijing, China) and 10 ng template DNA. PCR products were separated from impurities and genomic DNA by running in 2% agarose gels. The PCR bands were further purified using the AxyPrep DNA Gel Extraction Kit (Axygen Biosciences, Union City, CA, USA), and quantified using QuantiFluor-ST (Promega, USA) according to the manufacturer's protocols. Equimolar amounts of purified amplicons were pooled and paired ended sequenced (2 × 300) on an Illumina MiSeq platform (Illumina, San Diego, USA) according to the standard protocols of Majorbio Bio-Pharm Technology Co. Ltd. (Shanghai, China). The reads were de-multiplexed using the Illumina software and separate FASTQ files were generated for each specimen and deposited to the Sequence Read Archive NCBI under the BioProject accession PRJNA470548. Another public archive repository is available at Figshare doi: https://doi.org/10.6084/m9.figshare.7205102.

## Raw data processing

Raw data were processed according to the standard protocols provided by Majorbio Bio-Pharm Technology Co. Ltd. (Shanghai, China) as previously described (*Liu et al., 2018*; *Wang et al., 2018*). In short, raw sequencing data was first de-multiplexed. Sequence reads were then subjected to quality filtering utilizing Trimmomatic software (*Bolger,*

*Lohse & Usadel, 2014*) and were truncated at any site with a Phred score < 20 over a 50 bp-sized window. Barcode matching with the primer mismatch from 0 to 2 nucleotides was adopted and reads containing ambiguous characters were removed. After trimming, FLASh (Fast Length Adjustment of Short Read) (*Magoč & Salzberg, 2011*), a read pre-processing software, assembled and merged the paired-end reads from fragments and generated > 10 bp overlapped, with the dead match ratio of 0.2. Unassembled reads were discarded. From the 192 fecal samples sequenced, a total of 7,472,400 optimized V3-V4 tags of 16s rRNA gene sequences were generated (Table S1).

To unbiasedly compare all the samples at the same sequencing depth, the "sub.sample" command of mothur program (version1.30.1) (*Schloss et al., 2009*) was used for normalization to the smallest sample size. Chimera was detected and removed by UCHIME Algorithm. The effective reads were then sorted by cluster size and processed using Operational Taxonomic Units (OTUs) with 97% similarity cutoff UPARSE-OTU algorithm (implementing "cluster_otus" command) (*Edgar, 2013*) in USEARCH(v10)(UPARSE version 7.1). The taxonomy of each 16S rRNA gene sequence was analyzed by RDP Classifier algorithm (*Wang et al., 2007*) against the Silva (SSU128) 16S rRNA database (*Quast et al., 2012*) using confidence threshold of 70%. Each sequence was assigned the taxonomy by QIIME (*Caporaso et al., 2010*). The representative sequences were allocated phylogenetically down to the domain, phylum, class, order, family, and genus levels (Table S2). The relative abundance of a given taxonomic group was calculated as the percentage of assigned sequences over total sequences.

Within-sample diversity (alpha diversity), including Shannon index and observed species richness (Sobs), was obtained using the "summary.single" command of mothur program (version1.30.1) (*Schloss et al., 2009*). Between-sample diversity (beta diversity) was obtained by calculating weighted UniFrac distances between samples.

## Statistical and bioinformatics analyses
### Demographics and clinical sample comparisons

Kruskal–Wallis test and Wilcoxon rank-sum test were used to identify statistically significant differences in continuous variables, including gestational age, birth weight, age at diagnosis and length of hospitalization. The $\chi^2$, or Fisher's exact test was used to identify differences in gender composition. α level was considered 0.05 for all statistical tests. Other statistical analyses not involving microbiome 16s rRNA sequencing data were performed using the "*stats*" package in R (v.3.5.1).

### Microbiota and bioinformatics analyses

*Disease-related time interval definition.* Considering that the sampling and disease onset time for each patient were not identical, to illustrate the continuous longitudinal and repeated nature of the sampling and its relationship with onset and progression of diseases, we divided the sampling span into seven time intervals:
1. early post-partum(EPP): within 3 days after birth;
2. early pre-onset(EPO): from the end of EPP to at least four days before disease onset;

3. late pre-onset(LPO): from the end of EPO to the disease onset; for control group patients, the equivalent onset time is set at the 16th day of life, as is the average diagnosis age of NEC and LOS groups;
4. early disease(ED): the first third interval of the whole disease span; for the control group, the equivalent ED interval is from day 16 to discharge;
5. middle disease(MD): the middle third interval of disease span;
6. late disease(LD): the last third interval of disease span;
7. post disease(PD): from the end of disease to discharge time-point.

*Modeling strategies for comparisons.* To compare the dynamics of microbiota diversity and relative taxonomic abundance preceding the disease, we applied the EPP, EPO, LPO and ED interval among all patients into our model or comparisons.

*Diversity analyses.* Kruskal–Wallis tests were used to compare the differences in overall alpha diversity. The Mann–Whitney $U$ test was then applied to compare two adjacent time intervals. Differences in alpha diversity over time were analyzed by a two-way repeated measures ANOVA, with the time interval (EPP, EPO, LPO, ED, MD, LD, PD) as a within-subject factor and the group (NEC, LOS, control) as a between-subject factor. If more than one sample of a patient were collected within a time interval, the average of the α diversity indices was used as one data point.

*Taxonomy comparisons.* Zero-Inflated Beta Regression Model with Random Effects (ZIBR) and Linear Mixed-effects Model (LME) were used to test the association between OTU relative abundance and clinical covariates (diseases-related time intervals) for longitudinal microbiome data (*Chen & Li, 2016*). ZIBR and *nlme* (*Pinheiro et al., 2018*) R packages were utilized for each model. If more than one sample of a patients were collected within a time interval, the average of relative abundance of each genus was used.

### Scripts and figures archiving

Figures were generated with the "*ggpubr*" (*Kassambara, 2017*), "*ggplot2*" (*Wickham, 2016*) and "*ggsci*" (*Xiao, 2018*) packages using R (v.3.5.1). RScripts for analyses as well as input and output files are available at our GitHub repository.

## RESULTS

### Patients characteristics

From July 2013 to December 2014, a total of 130 preterm infants admitted to the neonatal intensive care unit (NICU) of Shanghai Children's Medical Center met the criteria of our study and a total of 1698 samples were collected. 192 fecal samples from 24 well-sampled preterm infants were sequenced. Four subsequently developed NEC (2 in stage IIA and 2 in stage IIB) and three developed LOS (2 with positive hemoculture of Klebsiella pneumoniae; the other one was diagnosed upon sepsis-related signs and symptoms, lab test of white blood cells > 20 cells/microL and her effective reaction to vancomycin). The remaining 17 served as matched controls (Fig. 1, Table S3). Fecal samples were collected between days 1 and 69 of life. Numbers of samples collected and interval of sampling varied among patients

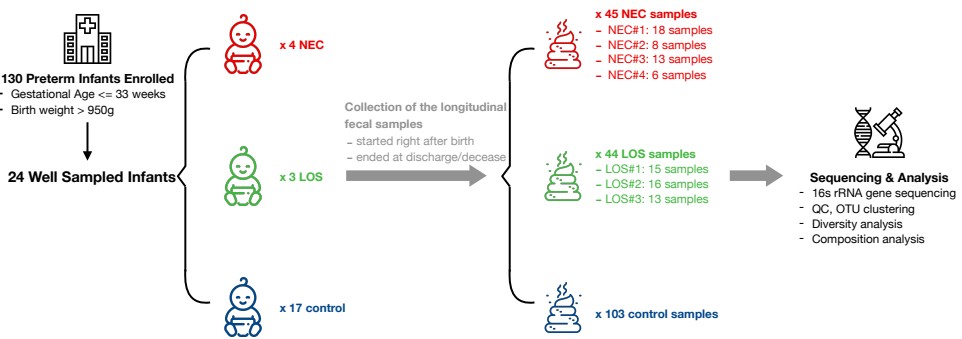

**Figure 1  Schematic of study design.** Longitudinal fecal samples were collected over from birth to decease or discharge from preterm infants in the NICU. Bacterial diversity and compositions were then character­ized. Image credit: Icons made by Freepik from http://www.flaticon.com.

**Table 1  Demographic characteristics of Preterm NEC, LOS and control groups.** There was no statistical differences in gestational age, birth weight, gender and age when diagnosed among NEC, LOS and Control group. The mean length of stay differs among the three groups, which is within our expectation because it takes longer time for NEC or LOS patients to recover.

|  | NEC ($N = 4$) | LOS ($N = 3$) | Control ($N = 17$) | Statistical test | *p* value |
|---|---|---|---|---|---|
| Gestational age (weeks) | 29(29–30) | 30(29–31) | 31(28–33) | Kruskal–Wallis test | 0.074 |
| Birth weight (g) | 1416.3 (773.4–2149.1) | 1141.7 (633.4–1649.9) | 1527.4 (1391.6–1663.1) | Kruskal–Wallis test | 0.111 |
| Gender |  |  |  | Fisher's exact test | 0.820 |
| Female | 3(75%) | 2(67%) | 9(53%) |  |  |
| Male | 1(25%) | 1(33%) | 8(47%) |  |  |
| Diagnosis age (days) | 16(11–19) | 16(10–22) | – | Wilcoxon rank-sum test | 0.629 |
| Length of stay(days) | 54.3 (13.5–95.0) | 60.0 (24.8–95.2) | 32.9 (26.3–39.5) | Kruskal–Wallis test | 0.046 |
| Number of samples | 45 | 44 | 103 | – | – |

but met our preset criteria of less than 7 days between sampling. The average number of sample collected for NEC, LOS and control patients was 11, 14 and six respectively. The number of samples per patient was higher in the NEC and LOS groups because the severity of the disease required longer hospitalization ($p = 0.046$).

All 24 infants profiled were delivered by Cesarean section, fed on infant formula and prescribed with prophylactic antibiotics regimen (cefotaxime, piperacillin-tazobactam and/or metronidazole) right after they were admitted to our NICU. No infant was prescribed probiotics during the study. There was no significant difference in gestational age ($p = 0.074$), birth weight ($p = 0.111$) or gender proportions ($p = 0.822$) among the three groups. The average age at diagnosis for both disease groups was 16 days and there was no statistical difference between the groups ($p = 0.629$) (Table 1). Therefore, we assigned day 16 to discharge as early disease interval, day 4–8 as early pre-onset interval and day 9–15 as late pre-onset interval for the control group (Table S4).

## Longitudinal Microbiome Diversity of NEC and LOS patients

To get an overview of gut microbiota in patients, we analyzed the microbial richness of the NEC and LOS patients over time. Similar to the control group, the case groups showed a

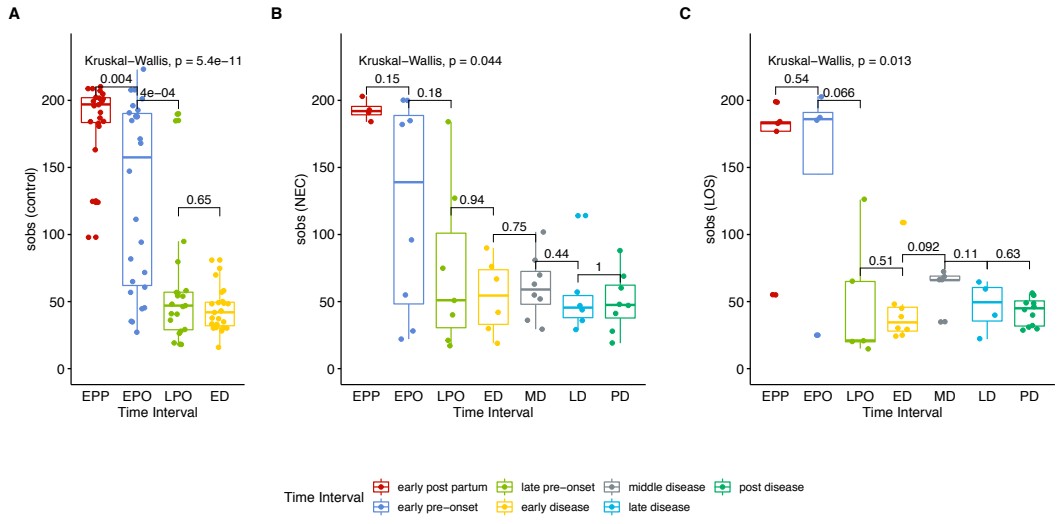

**Figure 2** **Trend in microbiome richness (Sobs) over time.** Shows microbial evenness trend in stools from cases and controls. (A) control group. (B) NEC group. (C) LOS group. Horizontal line shows median, box boundaries show 25th and 75th percentiles. Sobs index value of each sample is depicted as one dot. Indices are analyzed using the Kruskal–Wallis test followed by the Mann–Whitney $U$ test in comparisons between two adjacent intervals.

decreasing trend in observed species (Sobs) from early post-partum stage (EPP) to early disease (ED) stage (Fig. 2 (A) control group, $p < 0.01$; (B) NEC group, $p = 0.044$; (C) LOS group, $p = 0.013$; Dataset S1, Sheet "Sobs" two way RM ANOVA, $p < 0.0001$). The greatest decline in sobs was from early pre-onset (EPO) to late pre-onset (LPO). However, the decrease in the disease groups was less significant than the control group (control group $p = 0.0004$, NEC group $p = 0.18$, LOS group $p = 0.066$). The Sobs then stabilized from LPO onward with no significant difference between adjacent time intervals.

Next, we analyzed gut microbiome evenness over time. Similar to Sobs, the Shannon indices decreased significantly from the early post-partum (EPP) to early disease (ED) stage (Fig. 3A control group 2.768 to 1.004, $p = 0.04$; (B) NEC group, 3.141 to 0.578, $p = 0.01$; (C) LOS group, 2.641 to 0.470, $p = 0.01$).

Two way RM ANOVA showed significant Shannon index divergent among three groups before disease onset (Dataset S1, sheet "Shannon", EPP to ED, $p = 0.0017$). Moreover, during early disease stage, the Shannon indices were different among three groups (Fig. 4, facet early disease, $p = 0.0037$), suggesting that microbiota distortion may precede NEC and LOS onset. As diseases progressed, the NEC group differed significantly with the LOS group during middle disease interval but insignificantly during late disease interval (Fig. 4, facet middle disease, $p = 0.034$; facet late disease, $p = 0.750$). Upon alleviation of both diseases, the Shannon indices rose back to the early pre-onset levels (Fig. 3B) NEC group. early pre-onset at 1.925 vs. post disease at 1.320, $p = 0.79$; (C) LOS group, early pre-onset at 2.473 vs. post disease at 1.463, $p = 0.16$).

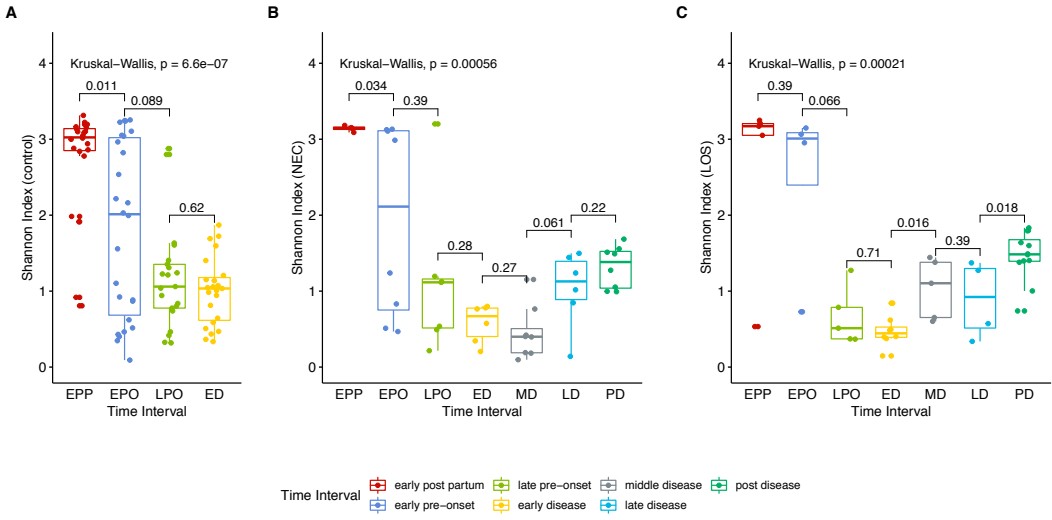

**Figure 3** **Post-partum microbiome evenness (Shannon diversity) trend in each group.** Shows microbial richness trend in stools from cases and controls. (A) control group. (B) NEC group. (C) LOS group. Horizontal line shows median, box boundaries show 25th and 75th percentiles. Shannon index value of each stool is depicted as one dot. $p = 0004$ for NEC and $p = 0.010$ for LOS from early preonset to early disease (two way RM ANOVA) indicating significantly discordant trends in bacterial diversity preceding disease onset. $p$ values between two adjacent intervals were calculated by the Mann–Whitney $U$ test.

## Kinetics of microbiome composition

To compare the beta-diversity of the three groups over time, we applied Principal Component Analysis (PCoA) to weighted UniFrac distance matrix (*Clarke, 1993*). Bacterial composition of three groups during early post-partum interval were the most similar compared with other time intervals, with the first principal coordinates accounted for 33.01% (Fig. 5A). Then beta diversity continued to separate from one another. The first principal coordinate one (PC1) increased from 33.01% at the early post-partum to 35.23% at the early pre-onset stage, 38.36% at the late pre-onset stage and eventually reaching 42.32% at the early disease stage (Figs. 5B to 5D). This continuous increase in beta-diversity suggested that the phylogenetic composition of the patients' microbiome started to deviate from the control group before the onset of diseases. As diseases progressed, the phylogenetic similarity between the NEC and the LOS disease groups diverged further and peaked at 59.53% in middle disease stage then came down gradually to 42.8% at post disease stage (Fig. 5E to Fig. 5G). This trend in phylogenetic dissimilarity suggested that the microbiome composition of the NEC and LOS patients might have deviated from normal even before the onset of diseases. Also, the further separation between the NEC and the LOS groups could be a result of different treatment strategies.

## Colonization trend at the genus level

In the analyses of intestinal microbiome alpha (Figs. 2–4) and beta diversity (Fig. 5), detectable differences were observed among the three groups, especially during the transition from the LPO to ED stage. This indicated that the microbiota assembly differences between the case groups and control group. To further investigate which

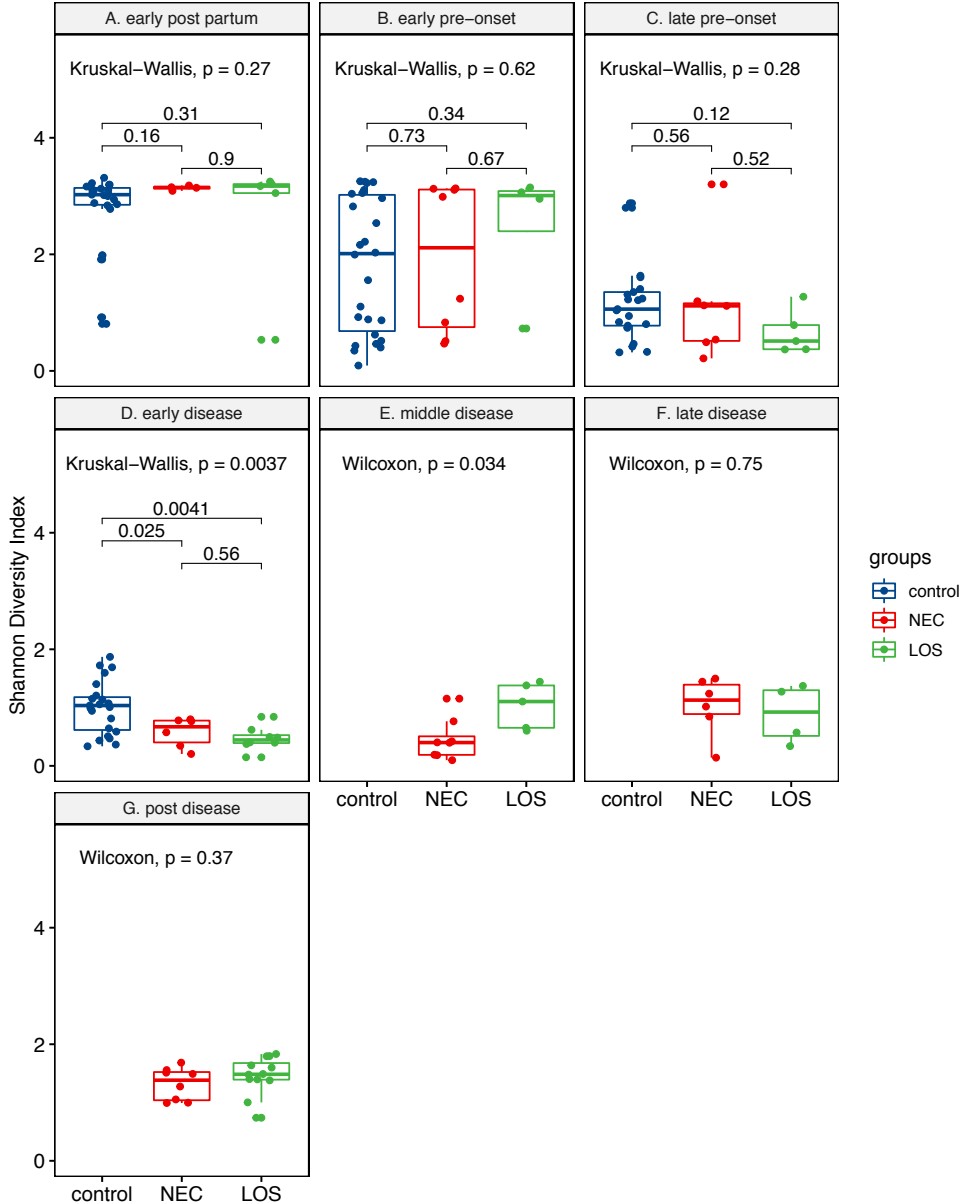

**Figure 4** **Post-partum microbiome evenness (Shannon diversity) in each time-interval.** Shows microbial richness trend in stools from cases and controls. (A) Control group. (B) NEC group. (C) LOS group. The horizontal line shows median, box boundaries show 25th and 75th percentiles. Shannon index value of each stool is depicted as one dot. *p* values among three groups were calculated by Kruskal–Wallis test. Comparisons between two groups were calculated by the Mann–Whitney test.

microbiota composition was correlated with the onset and/or progression of NEC and LOS, we tracked the longitudinal compositional changes in genera abundance. We filtered the genus of over 10% relative abundance among all samples and plotted relative abundance over time (Fig. 6).

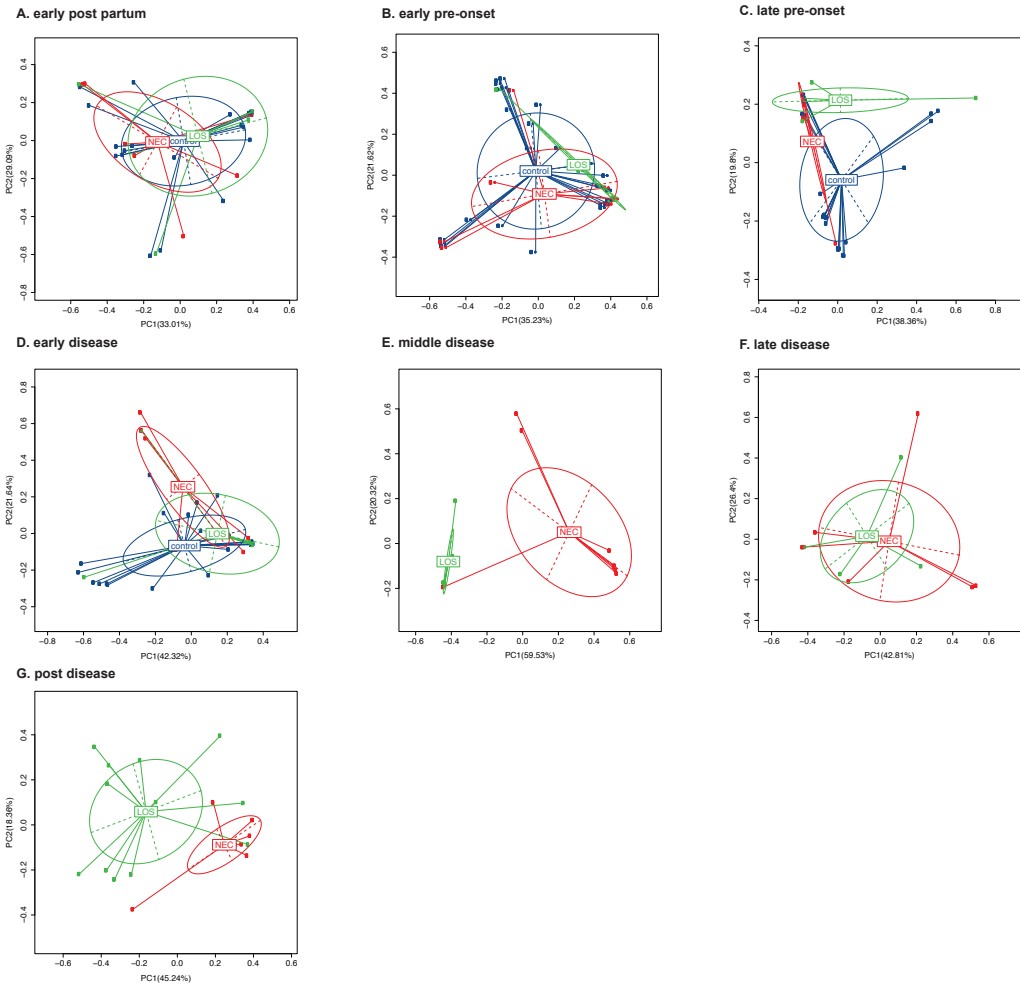

**Figure 5** **Beta diversity of the NEC, LOS and the control groups over time.** (A) Early post partum; (B) early pre-onset; (C) late pre-onset; (D) early disease; (E) middle disease; (F) late disease; (G) post disease. Beta diversity of samples is depicted by principal coordinates analysis 'PCoA' plot showing weighted UniFrac distance between samples. Each dot represents the microbiota of a single sample. Samples from the same group is represented by the same colors. Scatter plot shows principal coordinate 1 (PC1) versus principal coordinate 2 (PC2). Percentages shown are percentages of variation explained by the components. Samples that clustered closer together are considered to share a higher proportion of the phylogenetic tree and represented by a higher percentage in PC1.

At the early post-partum stage, all three groups showed high proportion of *Lactococcus*, *Bacillus* and *Pseudomonas*. However, ZIBR model the disease groups showed significantly higher OTUs that matched to *Bacillus* (NEC 15.05% and LOS 15.97% compared to 6.02% of control, $p = 0.032$) and *Solibacillus* (8.88% in NEC and 9.61% in LOS compared to 3.65% of control, $p = 0.047$) from the case groups (Dataset S2). Moreover, *Enterococcus* proportion (Fig. 6B, purple area) was much higher in LOS patients (20.72%) than the normal controls (6.66%, Fig. 6A, purple area) but almost absent in NEC patients (0.51%) (Fig. 6B). While all three groups showed increases in *Klebsiella* and *Escherichia-Shigella* and decrease in *Lactococcus* from EPP to ED, the rates of change were different among the

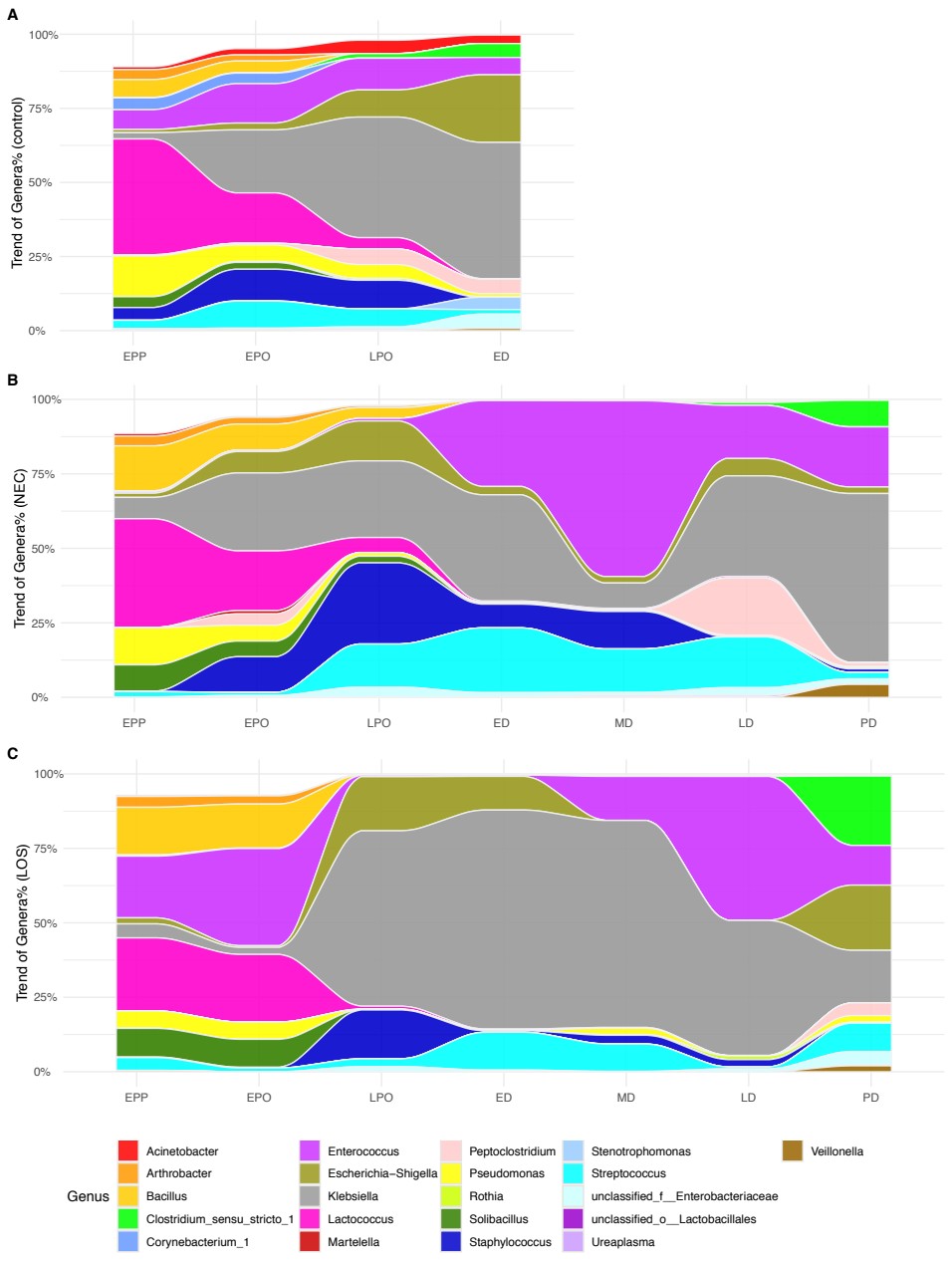

**Figure 6  Alluvial diagrams showing temporal development of microbiome in NEC, LOS and the control groups.** Genus of relative abundance over 10% were depicted. (A) Control group. (B) NEC group. (C) LOS group.

three groups. The LOS group exhibited the most drastic changes, with a rapidly increase of *Klebsiella* (from 4.71% to 58,90%), *Escherichia-Shigella* (from 2.02% to 18.16%) and *Streptococcus* (from 1.22% to 12.68%) (Fig. 6C). Together, these three genera accounted for almost 100% of all bacteria (Fig. 6C). In addition, *Lactococcus* decreased more rapidly than the other groups, from 24.54% at EPP to 0.94% before LPO (Fig. 6C magenta area).

Besides, the increase of *Klebsiella* was the most minimal in NEC patients (Fig. 6B grey area, from 7.17% at EPP to 35.63% at ED). Moreover, a rapid surge of *Enterococcus*, *Staphylococcus* and *Streptococcus* from EPO to ED was only observed in NEC patients (Fig. 6B, purple, dark and light blue area).

As NEC and LOS progressed with medical intervention, the genus in case groups underwent another round of drastic changes. Most notably, the fluctuation of *Enterococcus*, *Klebsiella*, *Staphylococcus* and *Peptoclostridium* during the disease stages (Figs. 6B and 6C stage ED to LD), which might be resultant from different healthcare strategies applied in two groups. Interestingly, as patients approached remission, the composition became more balanced and resembled more to that of the normal control, except for a higher level of *Clostridium*. In summary, relative to patients in the control group, we observed different patterns of temporal alterations in bacterial composition among NEC and LOS patients. Rapid changes in relative abundance of certain genera were revealed as early as early pre-onset of stages and were the most notable in LOS patients.

## DISCUSSION

In this pilot study, we intend to investigate the etiopathology of NEC and LOS in Chinese preterm infants from the perspective of intestinal microbiota. We profiled the gut microbiome of NEC and LOS preterm infants from birth to death or discharge. Some of our findings are similar to previous larger-scale studies. Mainly, infants who developed NEC or LOS exhibit a different gut microbiota colonization pattern relative to the controls. Case groups showed a decline in diversity, although to a different extent. Moreover, NEC and LOS infants' intestines were prone to harbor potential pathogens prior to and after disease onsets, such as *Enterococcus*, *Staphylococcus*, *Peptoclostridium* and *Streptococcus*. There were also findings unique to this study will be discussed in the following paragraphs.

To our knowledge, few studies have analyzed stool bacterial alpha diversity in preterm infants as early as three days after birth. Unexpectedly, within three days after birth (i.e., early post-partum interval), the bacterial diversity of all three groups was the highest compared to the following stages. At this point, we do not know if this high bacterial richness and evenness within three days of life are universal. More data, especially from other countries, are needed to support this finding. After three days, the microbial alpha diversity exhibited a declining trend in both disease groups and the control group. The number of colonized species (sobs index) during this interval, in line with previous works (*Mai et al., 2011*; *Mai et al., 2013*), remained similar before disease onset in both case and control groups, suggesting a minor role of bacterial richness in the disease onset. Besides, a rapid decline in alpha diversity during the pre-onset stages was observed. This could be resultant from the standardized antibiotic regimen right after admission into our NICU. However, previous studies showed that the pervasive effect of antibiotics in reducing richness and evenness arose only after 1 week to 2 months of administration (*DiGiulio et al., 2008*; *Dethlefsen & Relman, 2011*; *Fouhy et al., 2012*; *Greenwood et al., 2014*; *Tanaka et al., 2009*). Thus, more research is needed to identify if additional factor(s) is involved in this rapid decline.
The role of empiric prophylactic antibiotics in NEC or LOS is controversial. In animal models, antibiotics eliminating Gram-negative bacteria enhance gut function and diminish mucosal injury to the bowel thus preventing necrotizing enterocolitis or bacterial leakage into the bloodstream (*Carlisle et al., 2011*; *Jensen et al., 2013*; *Birck et al., 2015*). In clinical practices, broad-spectrum antibiotics (the most commonly prescribed medications in the NICU) are recommended to empirically prevent and treat both NEC and LOS (*Bury & Tudehope, 2001*; *Brook, 2008*; *Kimberlin et al., 2018*). However, antibiotics can further induce microbiome dysbiosis that may increase the risk of developing these diseases and exacerbate the severity (*Gibson, Crofts & Dantas, 2015*; *Kuppala et al., 2011*; *Martinez et al., 2017*; *Cantey et al., 2018*). Our results showed limited differences in bacterial diversity and composition between two case groups and the control group despite continuously antibiotics administration. Although our results are in line with the dysbiotic effect of antibiotics, there was not enough evidence to support whether antibiotics per se induced or prevented NEC and LOS. Further studies are needed to confirm the causative relationships.

Furthermore, microbiota beta-diversity, which measures the phylogenetic similarity, drifted away continuously among three groups before the onset of both diseases. These findings were inconsistent with a previous study where the microbiota of NEC patients were shown to be similar to that of the healthy controls at three days before onset (*Mai et al., 2011*). With regards to the LOS patients, it is also inconsistent with the previous study where similar microbiota diversity was observed in LOS patients during the disease and 72 h before onset (*Mai et al., 2013*). These discrepancies could be a result of differences in collection time points or differences in patients' demographics. Further studies are necessary to address these issues. As the diseases progressed, the beta-diversity of the NEC group and the LOS group separated further but converged again when diseases were alleviated. The exact cause of this divergence was not clear. It could be related to different treatment strategies or some intrinsic pathophysiology differences between the two diseases. Further studies should provide more insight.

In addition to bacterial diversity, we also tracked longitudinal changes in composition at the genus level by plotting the relative abundance over time. Overall, the control group exhibited more stable microbiota assembly, without drastic fluctuation in genus abundance and with less dominance of facultative anaerobes such as *Enterococcus* and *Staphylococcus* (*Gibson, Crofts & Dantas, 2015*; *La Rosa et al., 2014*; *Grier et al., 2017*). Based on our ZIBR model, an over-represented Bacillus and Solibacillus were detected during the pre-onset stages in case groups. However, both genera diminished after disease onset suggesting that the initial microbiota composition in preterms might contribute to their future health outcomes. Previous studies also observed a surge in Proteobacteria phyla (*Mai et al., 2013*; *Mai et al., 2011*) preceding LOS and NEC onset. In line with this, LOS patients in our cohort were also characterized by a higher abundance of *Klebsiella* in their intestinal communities. On the contrary, NEC infants presented overgrowth of *Streptococcus* and *Staphylococcus* (both belong to phyla *Firmicutes*) before disease onset. Further work is warranted to identify specific genera and trends in association with the onset of NEC and/or LOS.

Diarrhea is one of the typical symptoms in NEC patients and *Peptoclostridium* is conventionally regarded as a causative pathogen of hospital-acquired infectious diarrhea

(*Rodriguez et al., 2016*; *Pereira et al., 2016*). In our study, we identified a transient bloom of *Peptoclostridium* in late NEC stage that coincided with the diarrhea symptom, possibly explaining the mechanism of common diarrhea symptom in NEC patients. Moreover, mucosal-adhering bacteria such as *Enterococcus* and *Streptococcus* were highly represented in pediatric enterocolitis (*Normann et al., 2013*; *Zhou et al., 2016*). Consistent with this, NEC patients from our cohort exhibited a higher abundance of *Enterococcus* during disease stage.

In contrast, the composition of our LOS patient samples was very different from previous studies where *Enterobacteria* and *Staphylococcus* were identified as the most prevalent genera (*Stewart et al., 2017*; *Mai et al., 2013*). In our cohort of LOS patients, *Klebsiella* was the most dominated genus. LOS is frequently caused by organisms, mostly bacteria, that translocate from the intestinal tract to the bloodstream. Consistently, *Klebsiella* was detected in hemoculture in two out of two of our LOS patients (hemoculture was not performed for the third patient). In addition, *Klebsiella pneumoniae* is one of the most common causes of sepsis in preterm patients of our hospital (JL and LH personal observation), suggesting that the most dominant and eventually infectious bacteria may be more specific to the environment.

Another notable point in our cohort was almost absent *Bifidobacteria*, an anaerobe that can ferment milk oligosaccharides (*Gomez-Gallego et al., 2016*) and thus commonly detected among breastfed infants (*Murphy et al., 2017*). We speculate that this extremely low level in our cohort was due to the lack of breastfeeding in the sterile hospital environment, being nurtured in the sterile NICU environment, continuous administration of antibiotics or the combinations of the above. Although *Bifidobacteria* has been generally considered a probiotic that serves to protect neonates against necrotizing enterocolitis and systemic infection (*Nakayama et al., 2003*; *Khodayar-Pardo et al., 2014*; *Hermansson et al., 2019*), recent randomized controlled trials are showing paradoxical results (*Hays et al., 2016*; *Singh et al., 2019*). Further studies on the role of probiotics in optimizing preterm infants' microbiota should address their effectiveness in preventing NEC and LOS.

This study was limited to only one hospital in one specific region (Shanghai) in China so how far these findings can be extrapolated remains to be determined. In addition, our sample size was relatively small since both diseases are rare (*Neu & Walker, 2011*; *Cohen-Wolkowiez et al., 2009*). Among the 1,148 preterm infants admitted from July 2013 to December 2014, only five developed NEC and seven developed LOS. Nevertheless, this pilot study has provided essential information about NEC and LOS preterm patients within the Chinese population and serves as a starting point for future investigations into the etiology and pathogenesis of both diseases in the nation.

## CONCLUSIONS

In this longitudinal study, we used next generation-sequencing to profile the microbiota of 24 Chinese preterm infants from birth to discharge. Among them, four developed NEC and three developed LOS. To our knowledge, this is the first profiling of gut microbiota in NEC and LOS patients among the Asian population. Reduction in intestinal microbiota

diversity and divergence of phylogenetic similarity from the control infants over time associated with both NEC and LOS onset. Overgrowth of potentially pathogenic genus *Enterococcus*, *Streptococcus* and *Peptoclostridium* were observed in NEC cases while *Klebsiella* was recognized as the dominant genus in LOS cases. In summary, our findings suggest that both NEC and LOS are dynamic processes involving abnormal microbiota assembly. This study is a starting point for further studying of microbial factors involved in preterm-associated complications in China. Accumulation of more data within China and perhaps from neighboring countries will allow us to build microbial signatures that can assist early diagnosis and development of novel treatment.

## ACKNOWLEDGEMENTS

We sincerely thank all the patients and their families for participating in this study. We extend our thanks to the medical and research staffs of the Shanghai Children's Medical Center. We also thank Ka Ming Pang, Arin Nam and Christina Meyer for critical reviews of this manuscript.

### Funding

This work was supported by the National Natural Science Foundation of China (No. 81771630). The funders had no role in study design, data collection and analysis, decision to publish, or preparation of the manuscript.

### Grant Disclosures

The following grant information was disclosed by the authors:
National Natural Science Foundation of China: 81771630.

### Competing Interests

Jianhua Zhao is an employee of the Shanghai Majorbio Bio-Pharm Technology Co., Ltd.

### Author Contributions

- Jiayi Liu conceived and designed the experiments, participated in the clinical coordination, performed the experiments, managed the biological samples and the raw sequencing data, analyzed the data, contributed reagents/materials/analysis tools, maintained source code, prepared figures and/or tables, authored or reviewed drafts of the paper, approved the final draft.
- Yuqing Li enrolled the patients, collected and managed the samples, participated in the clinical coordination.
- Yi Feng, Liya Pan, Zhoulonglong Xie and Zhilong Yan collected the samples.
- Li Zhang and Mingkun Li analyzed the data.
- Jianhua Zhao prepared figures and/or tables.
- Jianhua Sun collected and managed the samples, participated in the clinical coordination.
- Li Hong conceived and designed the experiments, authored or reviewed drafts of the paper, approved the final draft.

## Human Ethics

The following information was supplied relating to ethical approvals (i.e., approving body and any reference numbers):

Joint Committee of Ethics of Shanghai Children's Medical Center, School of Medicine Shanghai Jiao Tong University approved this study (Approval Number SCMCIRB-K2013022).

## Data Availability

Liu, Jiayi; Sun, Jianhua; Li, Yuqing; Feng, Yi; Pan, Liya; Xie, Zhoulonglong; et al. (2019): 192 samples - microbial sequencing data for publishing ''Patterned progression of gut microbiota predisposes preterm infants to necrotizing enterocolitis and late-onset sepsis: data from a non-Western population''. figshare. Dataset. https://doi.org/10.6084/m9.figshare.7205102.v3.

RScripts for analyses as well as input and output files are available at GitHub: Available at https://github.com/jiayiliujiayi/NEC-LOS-microbiota_pattern_comparison.

## Supplemental Information

Supplemental information for this article can be found online at http://dx.doi.org/10.7717/peerj.7310#supplemental-information.

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
