# Peer review of "Patterned progression of gut microbiota associated with necrotizing enterocolitis and late onset sepsis in preterm infants: a prospective study in a Chinese neonatal intensive care unit"

_PeerJ, doi:10.7717/peerj.7310_

## Round 0.1 · original submission · Major Revisions

The reviewers have raised major concerns on your study. While one reviewer reported that the manuscript should be rejected, in light of the other comments put forward, we are offering you the opportunity to address their many concerns.

Please note that if you intend to re-submit the manuscript, it will be necessary to carefully address the many issues raised, including data re-analyses. Further, the quality of the language in your manuscript is poor, and will require considerable re-writing, and major editorial revision by someone fluent in English.

·

Basic reporting

Authors present a cohort of preterm babies followed by longitudinal stool sampling at NICU. Small subset of babies developed NEC (n=4) or LOS (n=3) which are then compared to healthy controls. The presentation of the data, bioinformatic and statistical methods used and English language are currently inadequate for publishing the manuscript.

The introduction is jumping between different topics and many citations are old and appeared in low impact journals (e.g. Matamoros et al.). There is abudance of recent work in infant gut microbiome which should be cited to introduce readers to the field. The importance of sentence on lines 97-100 is unclear. I suggest starting with couple of sentences on post-birth gut microbiome, briefly defining the conditions in focus (NEC and LOS), describing previous gut microbiome work on these diseases and leading up to the motivations and goals of the current study.

Lines 326-328: Authors are overselling their study

Experimental design

Authors use an outdated method for processing 16S rRNA gene sequecing data. Independent comparisons operating with mock community data have shown that uparse is prone to false positive OTUs (see e.g. PMID: 27822515). Authors should process the 16s amplicon data using an up-to-date method such as DADA2 (PMID: 27214047) or deblur (PMID: 28289731)

Validity of the findings

Figure 2 legend is missing and any information on statistical test used is missing even though p-values are reported on panel B. Panel B is also mostly unreadable since the relative abundance of most taxa is low and the bars are not scaled accordingly. In Methods, explain the statistical methods used and how do they account for longitudinality in the data.

Authors compare bacterial OTU counts between groups but this is not feasible since the groups differ by size and lacks any statistical assessment. I suggest measure bacterial richness (number of OTUs) or other measures of alpha-diversity per each stool sample and conducting statistical testing between groups. Longitudinal nature of the data (repeated measures from same subjects) needs to be taken in to account in selecting the statistical test used. Similarly, comparing mean relative abundances between groups is not adequate since it doesn't account for repeated measures. I suggest using mixed effects linear models or something similar to conduct statistical testing for relative abundances.

Additional comments

BioProjecgt PRJNA470548 was not found in NCBI SRA.

·

Basic reporting

A small scale study of NEC and LOS in preterm infants, showing the difference in basic metagenomes between controls and NEC/LOS. Clear and unambiguous, professional English used throughout. Sufficient field background/context provided. Professional article structure, figs, tables.

However, some novel data is first presented only in the discussion section.

Raw data (link to SRA) has been shared privately. However, a link to the SRA does not appear to be with the paper text. There needs to be a mention of the SRA ID in the text. Apologies if I missed it.

Experimental design

Very well done experimental design and research question. Methods sufficiently described. Sequencing depth (data generation) and analysis pipeline is appropriate to achieve aims of analysis.

However, results reveal that the design was underpowered for full 16S analysis across the two treatment and one control group. Thus, few significant results. Appropriately powered to generate a hypothesis, seems appropriate for a small journal such as PeerJ.

Validity of the findings

Largely negative/inconclusive results. Small (but significant) differences seen between case and control groups. Conclusions from the data are not overstated. Are reasonable, given the results.

Since NEC average onset was 16 (within the 14-21d range) and LOS was 12 (within the 7-14d range), why is LOS not showing a similar significant trend for the 7-14d range in a similar way to the 14-21d range shows significant differences in NEC?

Lines 285-287: Introducing new data in the discussion section. Why are these figures not reported in the results section?

Additional comments

Line 39: "dysbiotis" should be "dysbiosis"
Line 46: "rest17" should be "the rest (17)"
Line 67: delete "For" and start sentence with Preterm infants, who are prone...
Line 255: "less significant" -- the p value is > 0.05 so it is not a significant finding. Please correct the text.
Line 269: change "little" to "few"
Line 270: change "that" to "those"
Line 279: I believe you mean "7,472,400"
Line 317: "<Fig. S2" remove "<"


Figures:
S1a-c: Order of the legend is confusing. Please re-order the legend so that items in the legend from top to bottom are in order of days (i.e. 0-7 on top, and >28 on bottom)
Fig 2: Since PC1 and PC2 are such a high proportion, there is no need for S2b and S2c. S2a is sufficient to show the trend. Other figures are distracting.

·

Basic reporting

The English is not acceptable (see general comments for some additional information)

No P values are provided in the abstract and so there is no way of knowing if the reported diversity of taxa %s are important. The same largely applies for the main results.

Sometimes the authors report 3 NEC and 4 LOS, and other times they report 4 NEC and 3 LOS.

Experimental design

With only 24 infants and a total of 192 stool samples, this study falls well short of most studies published in this area over the past years. As an aside, many of these recent larger studies are not cited in the manuscript. Additionally, with only 4 NEC and 3 LOS infants, the study is greatly underpowered and fails to add substantially to the current literature.

It is not clear which samples were from the diaper or from the perianal skin surface. Furthermore, the latter represents a unique and none validated sample – is this representative of gut or stool? I also wonder about the ethics of collecting perianal samples using a spatula from extremely preterm infants.

With read lengths of ~500bp there is likely to be a high error rate in the data. Pat Schloss explains this perfectly in his excellent blog post - http://blog.mothur.org/2014/09/11/Why-such-a-large-distance-matrix/

Validity of the findings

How were the samples normalised? There is no mention of rarefaction or other?

No P values are reported for the alpha diversity or taxonomic comparisons, with the exception of the random P value on lines 234 and 235. For the second P value the authors report “weakly significant”, but this P value was 0.11 and is therefore not significant.

Additional comments

The English is generally poor throughout making the manuscript challenging to read. While I cannot go through the entire text line-by-line and improve the English, from the abstract alone here is a selection of some errors:
- line 14 I think the authors mean “the remaining 17 were..”, not “the rest17”.
- Sometimes abbreviations are used and other times they are wrote in full, e.g., line 50 Late-onset sepsis should be LOS as it has already been abbreviated, in the conclusion for both NEC and LOS
- Line 50-51 “…held the least diversified gut microbiota” is poor English, similarly line 52 “with the control group held the most diversified one” is also poor English.
- Line 54 “Both two groups”

It is not clear why sometimes rRNA is used and other times rDNA is used. The authors should be clear this is 16S rRNA gene sequencing and use this phrase throughout (e.g., in place of rDNA).

It is not stated what organism was cultured in the third LOS infant.

Centroids should be added to Figure 4, otherwise it is impossible to see what is going on.

---

## Round 0.2 · Minor Revisions

Thank you for your efforts in trying to address the many comments that were previously provided by the Reviewers. They have all commented that your revised version is markedly improved compared to your initial submission. Nonetheless, the Reviewers have highlighted a few lingering issues with your data analyses and interpretation that need to be addressed.

·

Basic reporting

-

Experimental design

-

Validity of the findings

-

Additional comments

Authors have addressed my previous concerns and the manuscript is improved greatly. I have one final comment on the revised manuscript:

lines 255-259: Authors are over-interpreting the increasing trend in the variance explained by principal coordinate 1. This increase only reflects the ability of the PCoA graph to display more variation in the data; i.e. the data is better represented by 2D graph. It does not necessarily tell anything about the separation between the groups. The separation between the groups can to be measured by PERMANOVA (adonis test) R-squared (variance explained by the grouping) or other similar tests. Note that adonis test is not able to account for repeated sampling.

·

Basic reporting

I believe this is an impressive level of re-do on the resubmission. I have no further issues with this article.

Experimental design

no comment

Validity of the findings

no comment

Additional comments

Congratulations on the vast improvement of this paper.

·

Basic reporting

The English is much improved

Experimental design

I am confused by the new analysis where authors have divided the sampling span into seven time intervals. It is not clear if repeated measures (i.e., samples from the same individual appear in this analysis)?

I am not clear if the LME adjusted for multiple comparisons?

Validity of the findings

no comment

Additional comments

The issue of repeated measures appears to be outstanding and will skew/inflate the P values.

Figs 2 and 3 compare changes over time within each group, but not between groups. Therefore, I am unable to see how the authors can make claims about about NEC and LOS showing differences to controls when this direct comparison was not performed. This is partially addressed in Fig 4, and it is interesting that the Shannon diversity is higher in controls in early disease, but one wonders if this simply reflects the use of antibiotics in the disease groups (which are known to reduce diversity).

My earlier point about adding centroids to the PCoA plots has not been addressed. Additionally, the authors should add the P values to these plots and inline with the above comments need to ensure they do not have repeated measures in these plots.

The authors have not stated which rarefaction level they used - this needs to be added to the main text.

I wonder if Fig 6 could be improved by making it clear which genera were significant and if they were higher in NEC, LOS, or controls

---

## Author Rebuttal · Round 0.2

14 Mar 2019

Dear Dr. José Derraik,

We sincerely thank the reviewers for their generous comments on our manuscript.

In response to the reviewers' comments and suggestions, we have reanalyzed our data and rewritten most parts of our manuscript. Here are the major revisions:

• English language improvement, in response to Reviewer 1 and 3.
• Utilization of the Zero-Inflated Beta Random Effect model to illustrate the longitudinal natures of our data, in response to Reviewer 1; utilization of appropriate statistical tests (including two way RM ANOVA) and reported all p-values, in response to Reviewer 3.
• Because the disease onset time for each patient was not the same, we reset the analysis time intervals as: early post-partum, early pre-onset, late pre-onset, early disease, middle disease, late disease, and post disease, to better reflect the condition of the patients.
• In response to Reviewer 1,2, and 3, we updated all figures plotted by R to show explicit results.

On behalf of all the authors, I am submitting our new version of the manuscript for your assessment.

Enclosed please find our point-to-point responses to previous reviews. We believe our manuscript has been substantially improved and would be appropriate to publish in PeerJ.

Thank you very much again for your considerations.

Yours Sincerely,
Li Hong

On behalf of all authors.

# Responses to Reviewers

Minor revisions, if not explained, are applied to mostly for shorter manuscript and for minor modification along with revisions described in this document. This document includes our responses to reviewers' comments:

\* The reviewers' comments are in italic font. Our responses are in regular font and embedded in dotted borders.

# Reviewer1 (Tommi Vatanen)

## *Basic reporting*

*Authors present a cohort of preterm babies followed by longitudinal stool sampling at NICU. Small subset of babies developed NEC (n=4) or LOS (n=3) which are then compared to healthy controls.*

> **Response:**
> We acknowledge the limited sample size in our cohort. To our knowledge, this is the first study profiling NEC and LOS patients among the Asian population. So our data serves as a starting point for further studies in relevant fields.

*The presentation of the data, bioinformatic and statistical methods used and English language are currently inadequate for publishing the manuscript.*

> **Response**:
> We reanalyzed our data utilized Zero-Inflated Beta Random Effect model and other appropriate statistical methods including two way RM ANOVA. So we came up with more robust and reliable results, and the data illustration has been significantly improved.
> We rewrote the whole manuscript and got editing help from three of Jiayi Liu's previous lab mates at City of Hope National Medical Center with full professional proficiency in English. So we believe that our manuscript is now readable and understandable.

*The introduction is jumping between different topics and many citations are old and appeared in low impact journals (e.g. Matamoros et al.). There is abudance of recent work in infant gut microbiome which should be cited to introduce readers to the field. The importance of sentence on lines 97-100 is unclear. I suggest starting with couple of sentences on post-birth gut microbiome, briefly defining the conditions in focus (NEC and LOS), describing previous gut microbiome work on these diseases and leading up to the motivations and goals of the current study.*

> **Response**:
> We sincerely appreciate your suggestion and rewrote the introduction part. In detail:
>   1. We started by introducing microbiota and its role in human health (line 38-44).

2. Then we briefly reviewed how temporal microbiota in children would lead to (line 45-51), followed by the microbiota characteristics in preterm infants(line 52-56).
3. We described previous works focusing on NEC and LOS (line 57-65 and line 66-74, respectively).
4. Lastly, it came up with our motivations and goals of our study(line 75-87).

*Lines 326-328: Authors are overselling their study*

**Response**:

In our new manuscript, we have deleted this part.

We agree that we did oversell our story in the original description. Thank you very much for pointing this out. We apologize for overselling the story thus conveying misleading information.

### Experimental design

*Authors use an outdated method for processing 16S rRNA gene sequecing data. Independent comparisons operating with mock community data have shown that uparse is prone to false positive OTUs (see e.g. PMID: 27822515). Authors should process the 16s amplicon data using an up-to-date method such as DADA2 (PMID: 27214047) or deblur (PMID: 28289731)*

**Response**:

We sincerely appreciate your providing us with two updated DADA2 and deblur, which represents the state-of-the-art in the field of 16s data processing. For us, these two pipelines are entirely new to our team, so it still needs our time and efforts to optimize before we could get solid, reliable and reproducible results for publishing.

The previous study showed that DADA2 and UPARSE are both capable of discriminating samples by treatment, leading to similar biological conclusions (PMID: 28903732).

Taken together, we keep using our previous pipelines. However, in the near future, we will surely use DADA2 as our new pipeline to publish. We apologize for not using the pipelines mentioned in your comments.

### Validity of the findings

*Figure 2 legend is missing and any information on statistical test used is missing even though p-values are reported on panel B. Panel B is also mostly unreadable since the relative abundance of most taxa is low and the bars are not scaled accordingly. In Methods, explain the statistical methods used and how do they account for longitudinality in the data.*

**Response**:
We have reanalyzed our data to show their longitudinal nature. (Discussed below)

*Authors compare bacterial OTU counts between groups but this is not feasible since the groups differ by size and lacks any statistical assessment. I suggest measure bacterial richness (number of OTUs) or other measures of alpha-diversity per each stool sample and conducting statistical testing between groups.*

**Response**:
Thank you so much for your suggestions. We discarded the comparisons in merely OTU counts. Instead, we first reset the analysis time intervals as: early post-partum, early pre-onset, late pre-onset, early disease, middle disease, late disease, and post disease. (Because the disease onset time for each patient is not on the same day of life, it is not wise to analyze on a weekly scale. )(Detailed description is in line 174- 187). Then we calculate the sobs and Shannon diversity indices of each group in each interval, followed by statistical tests.

*Longitudinal nature of the data (repeated measures from same subjects) needs to be taken in to account in selecting the statistical test used. Similarly, comparing mean relative abundances between groups is not adequate since it doesn't account for repeated measures. I suggest using mixed effects linear models or something similar to conduct statistical testing for relative abundances.*

**Response**:

Yes, we agree that using mean taxa% is far from an adequate analysis.

In our new analysis, we use the Zero-Inflated Beta Random Effect model to illustrate the longitudinal nature of our data, in details:

1. We divided the whole sampling span into seven analysis intervals: the analysis time interval as: early post-partum, early pre-onset, late pre-onset, early disease, middle disease, late disease, and post disease(Detailed description is in line 174- 187).
2. We set "Treat" (i.e., NEC, LOS, control) , "Time" (as mentioned above) and "subjects" (24 patients identified by 24 unique IDs) as covariates and applied into the model.
3. We got the p-value of each OTU and assigned the OTUs to the genus.

From the model, we detected OTUs with significant differences which matched to *Bacillus* and *Solibacillus.* Back to our genus trend (Fig 6), we found out the most significant difference in these two genera. Thus we speculate the initial microbiota compositions may contribute to the later health outcomes in preterm infants.

For your perusal, the source code for our modeling is available in our GitHub repository.

Besides, we also did mixed effects linear mode and got similar results (source code data were shown along with the ZIBR model at the GitHub repository). So we illustrated the ZIBR results. Please let us know if we should present the other model or both.

For diversity indices, we also utilized appropriate statistical tests (including two way RM ANOVA) and reported all significant results.

For relative abundance, we used an alluvial plot (Fig6) to illustrate the trend of abundance changes over time and compare the abundance within each analysis interval to show their longitudinal natures.

### *Comments for the author*
*BioProjecgt PRJNA470548 was not found in NCBI SRA.*

**Response**:
As for our raw data, we've uploaded to a public platform at figshare and its doi is 10.6084/m9.figshare.7205102. And the data at SRA(SRX4056077) has been set "private until paper is published" at SRA, so it is not public for now and our apologies for the inconveniences.

# Reviewer2 (William Schierding)

## *Basic reporting*

*A small scale study of NEC and LOS in preterm infants, showing the difference in basic metagenomes between controls and NEC/LOS. Clear and unambiguous, professional English used throughout. Sufficient field background/context provided. Professional article structure, figs, tables.*

> **Response**:
> We appreciate your comments!

*However, some novel data is first presented only in the discussion section.*

> **Response**:
> We have moved the data from discussion to the results section.

*Raw data (link to SRA) has been shared privately. However, a link to the SRA does not appear to be with the paper text. There needs to be a mention of the SRA ID in the text. Apologies if I missed it.*

> **Response**:
> As for our raw data, we've uploaded to a public platform at figshare and its doi is 10.6084/m9.figshare.7205102. And the data at SRA(SRX4056077) has been set "private until paper is published" at SRA, so it is not public for now and our apologies for the inconveniences.

## *Experimental design*

*Very well done experimental design and research question. Methods sufficiently described. Sequencing depth (data generation) and analysis pipeline is appropriate to achieve aims of analysis.*

> **Response**:
> Thank you so much for valuing our works!

*However, results reveal that the design was underpowered for full 16S analysis across the two treatment and one control group. Thus, few significant results. Appropriately powered to generate a hypothesis, seems appropriate for a small journal such as PeerJ.*

> **Response**:
> We have enriched our analysis from the following perspective:
> - We compare the longitudinal diversity indices using two way RM ANOVA.
> - With the [Zero-Inflated Beta Random Effect model](), we illustrated the longitudinal natures of our data. From the model, *Bacillus* and *Solibacillus* are significantly different genus among the three groups.
> - We plotted an alluvial plot (Fig6) to show the chronological trend in genus abundance.

### *Validity of the findings*
*Largely negative/inconclusive results. Small (but significant) differences seen between case and control groups. Conclusions from the data are not overstated. Are reasonable, given the results.*

> **Response**:
> Thank you for your comments! With our updated analyses, our findings are more reliable and solid than before.

*Since NEC average onset was 16 (within the 14-21d range) and LOS was 12 (within the 7-14d range), why is LOS not showing a similar significant trend for the 7-14d range in a similar way to the 14-21d range shows significant differences in NEC?*

> **Response**:
> We've made a mistake here that the average onset for LOS was also 16.
>
> In addition, we set time intervals for each patient to better analyze our data (because each patient has different onset timepoints, it is not fair to plot PCoA on a weekly basis). The intervals are: early post-partum, early pre-onset, late pre-onset, early disease, middle disease, late disease, and post disease (Detailed description is in line 174- 187).

*Lines 285-287: Introducing new data in the discussion section. Why are these figures not reported in the results section?*

> **Response**:
> We have moved the data from discussion to results section.

### Comments for the author
*Line 39: "dysbiotis" should be "dysbiosis"*
*Line 46: "rest17" should be "the rest (17)"*
*Line 67: delete "For" and start sentence with Preterm infants, who are prone...*
*Line 255: "less significant" -- the p value is > 0.05 so it is not a significant finding. Please correct the text.*
*Line 269: change "little" to "few"*
*Line 270: change "that" to "those"*
*Line 279: I believe you mean "7,472,400"*
*Line 317: "<Fig. S2" remove "<"*

**Response**:
We accept all the amendment suggestions.

*Figures:*
*S1a-c: Order of the legend is confusing. Please re-order the legend so that items in the legend from top to bottom are in order of days (i.e. 0-7 on top, and >28 on bottom)*
*Fig 2: Since PC1 and PC2 are such a high proportion, there is no need for S2b and S2c. S2a is sufficient to show the trend. Other figures are distracting.*

**Response**:
We have updated our figures plotted using R with more clear illustrations.

# Reviewer3 (Christopher Stewart)

## *Basic reporting*

*The English is not acceptable (see general comments for some additional information)*

**Response**:
We rewrote the whole manuscript and got editing help from three of Jiayi Liu's previous lab mates at City of Hope National Medical Center with full professional proficiency in English. So we believe that our manuscript is now readable and understandable.

*No P values are provided in the abstract and so there is no way of knowing if the reported diversity of taxa %s are important. The same largely applies for the main results.*

**Response**:
We amended the reported taxa% and their p-values in the abstract. In the results, the diversity index values, the percentage of primary coordinates, relative abundances were clearly reported with p-values followed.

*Sometimes the authors report 3 NEC and 4 LOS, and other times they report 4 NEC and 3 LOS.*

**Response**:
Thank you for pointing this out. We have amended all the misleading numbers throughout our manuscript. The correct version is "4 NEC and 3 LOS patients", for your perusal.

## *Experimental design*

*With only 24 infants and a total of 192 stool samples, this study falls well short of most studies published in this area over the past years.*

**Response**:
We acknowledge the limited sample size in our cohort. To our knowledge, this is the first study profiling NEC and LOS patients among the Asian population. So our data serves as a starting point for further studies in relevant fields.

*As an aside, many of these recent larger studies are not cited in the manuscript. Additionally, with only 4 NEC and 3 LOS infants, the study is greatly underpowered and fails to add substantially to the current literature.*

**Response**:
In the new version of our manuscript, we updated our citations with larger studies, including the TEDDY study (PMID: 30356187; PMID: 30356187) and other cohorts (PMID: 28701177; PMID: 26969089).

Although our sample size is relatively underpowered, this study is the first profiling gut microbiota in NEC and LOS patients in Chinese population (to our knowledge, our apologies if we're wrong ). Also, the high rate of preterm birth in China increases the patient population at risk of developing NEC/LOS.

Therefore, it has provided essential information in Chinese population and serves as a starting point for further understanding the etiology and pathogenesis of both diseases in the nation.

*It is not clear which samples were from the diaper or from the perianal skin surface. Furthermore, the latter represents a unique and none validated sample – is this representative of gut or stool? I also wonder about the ethics of collecting perianal samples using a spatula from extremely preterm infants.*

**Response**:
We checked with the personnel who collected samples during the collection time and confirmed that: all the samples are collected from the diaper within 30 minutes after defecation. Therefore, we amended the time after defecation and deleted the "perianal skin" part in the manuscript, to eliminate the ethical concerns.

*With read lengths of ~500bp there is likely to be a high error rate in the data. Pat Schloss explains this perfectly in his excellent blog post - http://blog.mothur.org/2014/09/11/Why-such-a-large-distance-matrix/*

**Response**:
For our ~500 bp read length: We used primers 338F and 806R used (with the supposedly ~460bp read length) and illumina MiSeq (PE300) sequencing

platform so that we could get >100 bp redundancies, which could satisfy our QC standards. The procedures of QC, assembling and merging reads would allow us to control the qualities of bases and reduce error rates.

### *Validity of the findings*

*How were the samples normalised? There is no mention of rarefaction or other?*

**Response**:

We randomly subsampled to the lowest number of sequences produced from any sample to normalize our OTU dataset (PMID: 15592412) and we attached the rarefaction curve for your perusal. Please let us know if it should be shown or omit in the manuscript.

[Figure]

*No P values are reported for the alpha diversity or taxonomic comparisons, with the exception of the random P value on lines 234 and 235.*

> **Response**:
> In our updated manuscript, p-values are reported for every comparison in diversity indices both in text and figure descriptions (Fig 2, 3, &4). p-values for taxonomic comparisons are all mentioned in the text.
> We apologize for unprofessional data illustrations previously.

*For the second P value the authors report "weakly significant", but this P value was 0.11 and is therefore not significant.*

> **Response**:
> We agree with the reviewer's opinion and have deleted this part. Apologies for our unprofessional descriptions.

*Comments for the author*

*The English is generally poor throughout making the manuscript challenging to read. While I cannot go through the entire text line-by-line and improve the English, from the abstract alone here is a selection of some errors:*
*- line 14 I think the authors mean "the remaining 17 were..", not "the rest17".*
*- Sometimes abbreviations are used and other times they are wrote in full, e.g., line 50 Late-onset sepsis should be LOS as it has already been abbreviated, in the conclusion for both NEC and LOS*
*- Line 50-51 "…held the least diversified gut microbiota" is poor English, similarly line 52 "with the control group held the most diversified one" is also poor English.*
*- Line 54 "Both two groups"*

> **Response**:
> We rewrote the whole manuscript and got editing help from three of Jiayi Liu's previous lab mate at City of Hope National Medical Center with full professional proficiency in English. So we believe that our manuscript is now readable and understandable.

*It is not clear why sometimes rRNA is used and other times rDNA is used. The authors should be clear this is 16S rRNA gene sequencing and use this phrase throughout (e.g., in place of rDNA).*

> **Response**:
> We changed all wrongly expressed terms into "16S rRNA gene sequencing" and use this phrase throughout. Thank you so much for pointing this out.

*It is not stated what organism was cultured in the third LOS infant.*

**Response**:
For the third late-onset sepsis patient, the following chief complaints and past medical history explained why she is grouped in the LOS group:
The patient presented with constant fever, tachypnea, poor feeding and white blood cells >20 cells/microL; she didn't react well with empirical antibiotics usage for more than 7 days. Thus, according to the principles of beneficence, she was given vancomycin before the result of hemoculture could be available.  After receiving vancomycin,  the symptoms alleviated, CBC parameters went normal so that her diagnosis was traced back as"Late-onset Sepsis." This explanation part is included in the correspondent table (Table S3).    *This is the translation version of the medical records and prescription charter. We apologize if anything is confusing. We would like to provide her whole medical records if necessary.

*Centroids should be added to Figure 4, otherwise it is impossible to see what is going on.*

**Response**:
We've updated our figures and discarded Figure 4. Instead, Figure 5 serves as the original Figure 4.

---

## Round 0.3 · accepted · Accept

Thank you for carefully addressing the reviewers' comments.

·

Basic reporting

Authors have addressed my comments.

Experimental design

No comments.

Validity of the findings

No comments.

Additional comments

No comments.